# A Molecular Systematics and Taxonomy Research on *Trechispora* (Hydnodontaceae, Trechisporales): Concentrating on Three New *Trechispora* Species from East Asia

**DOI:** 10.3390/jof8101020

**Published:** 2022-09-27

**Authors:** Kaiyue Luo, Changlin Zhao

**Affiliations:** 1Yunnan Key Laboratory of Plateau Wetland Conservation, Restoration and Ecological Services, Southwest Forestry University, Kunming 650224, China; 2College of Biodiversity Conservation, Southwest Forestry University, Kunming 650224, China; 3Key Laboratory for Forest Resources Conservation and Utilization in the Southwest Mountains of China, Ministry of Education, Southwest Forestry University, Kunming 650224, China; 4Yunnan Key Laboratory for Fungal Diversity and Green Development, Kunming Institute of Botany, Chinese Academy of Sciences, Kunming 650201, China

**Keywords:** fungal diversity, morphology, southwest China, subtropical region, wood-inhabiting fungi

## Abstract

*Trechispora* are an important genus of wood-inhabiting fungi that have the ability to decompose rotten wood in the forest ecosystem. In this study, we reported three new species of *Trechispora*: *T. murina*, *T. odontioidea*, *T. olivacea* from a subtropical region of Yunnan Province, China. Species descriptions were based on a combination of morphological features and phylogenetic analyses of the ITS and LSU region of nuclear ribosomal DNA. *Trechispora murina* is characterized by the resupinate basidiomata, grandinioid hymenial surface with a greyish tint, monomitic hyphal system and ellipsoid, thick-walled, ornamented basidiospores; *T. odontioidea* has an odontioid hymenial surface with cylindrical to conical, blunt aculei and subglobose to globose, colorless, slightly thick-walled, ornamented basidiospores; *T. olivacea* has a farinaceous hymenial surface with olivaceous tint, basidia clavate and thick-walled, ornamented, broadly ellipsoid to globose basidiospores. Sequences of the ITS and nLSU rDNA markers of the studied samples were generated, and phylogenetic analyses were performed with maximum likelihood, maximum parsimony, and Bayesian inference methods. After a series of phylogenetic analyses, the 5.8S+nLSU dataset was constructed to test the phylogenetic relationship of *Trechispora* with other genera of Hydnodontaceae. The ITS dataset was used to evaluate the phylogenetic relationship of the three new species with other species of *Trechispora*. Using ITS phylogeny, the new species *T. murina* was retrieved as a sister to *T. bambusicola* with moderate supports; *T. odontioidea* formed a single lineage and then grouped with *T. fimbriata* and *T. nivea*; while *T. olivacea* formed a monophyletic lineage with *T. farinacea*, *T. hondurensis,* and *T. mollis*.

## 1. Introduction

Fungi form an essential branch of the tree of life, inferred from the important relationship with animals and plants [1], and it drives the carbon cycling in forest soils, mediate mineral nutrition of plants, and alleviates carbon limitations of other soil organisms as the decomposers and mutualists of plants and animals being the fundamental ecological roles [2]. Inferred from growing on a variety of the boreal, temperate, subtropical, and tropical divers vegetations, wood-inhabiting fungi have a rich diversity [3,4,5,6,7,8,9,10,11,12,13]. *Trechispora* P. Karst. belongs to Trechisporales, a small but strongly supported order in Agaricomycotina [14,15]. *Trechispora* (Hydnodontaceae Jülich) typified by *T. onusta* P. Karst., which is characterized by resupinate to effused basidiomata; a smooth to hydnoid to poroid hymenophore; ampullaceous septa; short cylindric basidia; and smooth to verrucose or aculeate basidiospores [5,16]. Currently, MycoBank and Index Fungorum have registered 121 specific and intraspecific names in *Trechispora*. About 60 species are currently accepted in *Trechispora* worldwide [5,17,18,19,20,21,22,23,24,25,26,27], of which 18 species of the genus have been found in China [28,29,30,31,32,33,34].

The high phylogenetic diversity on the corticioid Agaricomycetes based on two genes, 5.8S and 28S in which nine taxa of *Trechispora* nested into trechisporoid clade [35]. The molecular systematics suggested that *Trechispora* belonged to Hydnodontaceae and was related to genera *Brevicellicium* K.H. Larss. & Hjortstam, *Porpomyces* Jülich, *Sistotremastrum* J. Erikss., and *Subulicystidium* Parmasto [36], the similar morphological characters of *Trechispora* to these genera are basidiomata resupinate, hyphal system monomitic, cystidia absent [5,37]. The phylogeny of Trechisporales was inferred from a combined ITS-nLSU sequences, which revealed that several related genera *Porpomyces*, *Scytinopogon* Singer, and *Trechispora* grouped closely together and nested within Hydnodontaceae [38].

Based on the ITS and nLSU datasets, the phylogenetic study of *Trechispora* reports two new *Trechispora* species: *T. cyatheae* Ordynets, Langer & K.H. Larss. and *T. echinocristallina* Ordynets, Langer & K.H. Larss., which were found in La Réunion Island [24]. Recently, a new species of *Trechispora* has been reported from North America and China [26,33,34].

During the investigations of the corticioid fungi, Yunnan Province, China, we collected three fungal taxa, which could not be assigned to any described species within Hydnodontaceae. We present morphological and molecular phylogenetic evidence that support them as the three new species in *Trechispora*.

## 2. Materials and Methods

### 2.1. Sample Collection and Herbarium Specimen Preparation

Fresh fruiting bodies of the fungi growing on fallen angiosperm branches were collected in 2019 from the Honghe and Wenshan of Yunnan Province, China. The samples were photographed in situ and macroscopic details were recorded. Field photographs were taken by a Jianeng 80D camera (Tokyo, Japan). All photographs were focus-stacked and merged using Helicon Focus Pro 7.7.5 software. Once the macroscopic details were recorded, the specimens were transported to a field station where the specimens were dried on an electronic food dryer at 45 °C. Once dried, the specimens were labeled and sealed in envelopes and plastic bags. The dried specimens were deposited in the herbarium of the Southwest Forestry University (SWFC), Kunming, Yunnan Province, China.

### 2.2. Morphology

The macromorphological descriptions were based on field notes and photos captured in the field and laboratory. Color, texture, taste and odor of fruit bodies were mostly based on the authors’ field trip investigations. Color terminology follows Kornerup and Wanscher [39]. All materials were examined under a Nikon 80i microscope (Nikon Corporation, Tokyo, Japan). Drawings were made with the aid of a drawing tube. The measurements and drawings were made from slide preparations stained with cotton blue (0.1 mg aniline blue dissolved in 60 g pure lactic acid), Melzer’s reagent (1.5 g potassium iodide, 0.5 g crystalline iodine, 22 g chloral hydrate, aq. dest. 20 mL), and 5% potassium hydroxide. Spores were measured from the sections of the basidiomata and when presenting spore size data, 5% of the measurements excluded from each end of the range are shown in parentheses [40]. The following abbreviations were used: KOH = 5% potassium hydroxide water solution, CB = cotton clue, CB– = acyanophilous, IKI = Melzer’s reagent, IKI– = both inamyloid and indextrinoid, L = means spore length (arithmetic average for all spores), W = means spore width (arithmetic average for all spores), Q = variation in the L/W ratios between the specimens studied, and n = a/b ((a) number of spores were measured in total, coming from (b) number of specimen).

### 2.3. Molecular Phylogeny

The CTAB rapid plant genome extraction kit-DN14 (Aidlab Biotechnologies Co., Ltd., Beijing, China) was used to obtain genomic DNA from the dried specimens following the manufacturer’s instructions [41]. The nuclear ribosomal ITS region was amplified with the primers ITS5 and ITS4 [42]. The nuclear nLSU region was amplified with the primer pairs LR0R and LR7 (http://lutzonilab.org/nuclear-ribosomal-dna/, accessed on 7 June 2019). The PCR procedure used for ITS was as follows: initial denaturation at 95 °C for 3 min, followed by 35 cycles at 94 °C for 40 s, 58 °C for 45 s, and 72 °C for 1 min, and a final extension of 72 °C for 10 min. The PCR procedure used for nLSU was as follows: initial denaturation at 94 °C for 1 min, followed by 35 cycles at 94 °C for 30 s, 48 °C for 1 min, and 72 °C for 1.5 min, and a final extension of 72 °C for 10 min. The PCR products were purified and sequenced at Kunming Tsingke Biological Technology Limited Company (Kunming, Yunnan Province, China). All the newly generated sequences were deposited in NCBI GenBank (https://www.ncbi.nlm.nih.gov/genbank/, accessed on 28 November 2021) (Table 1).

The sequences and alignment were adjusted manually using AliView version 1.27 [52]. The datasets were aligned with Mesquite version 3.51. The 5.8S+nLSU sequences dataset was used to know the phylogenetic relationship of the three new species in *Trechispora* and related genera, and the ITS dataset was used to evaluate the phylogenetic relationships of the new species with known species of the genus. Sequences of *Porpomyces mucidus* (Pers.) that Jülich and *P. submucidus* F. Wu & C.L. Zhao retrieved from GenBank were used as the outgroup for the 5.8S+nLSU analysis (Figure 1) [34], and sequences of *Fibrodontia alba* that Yurchenko & Sheng H. Wu and *F. brevidens* (Pat.) Hjortstam & Ryvarden retrieved from GenBank were used as the outgroup for the ITS analysis (Figure 2) [24,34].

The three combined datasets were analyzed using maximum parsimony (MP), maximum likelihood (ML), and Bayesian inference (BI), according to Zhao and Wu [41]. Maximum parsimony analyses were constructed using PAUP* version 4.0b10 [53]. All characters were equally weighted and gaps were treated as missing data. Trees were inferred using the heuristic search option with TBR branch swapping and 1000 random sequence additions. Max trees were set to 5000, branches of zero length were collapsed, and all parsimonious trees were saved. Clade robustness was assessed using bootstrap (BT) analysis with 1000 replicates [54]. Descriptive tree statistics—tree length (TL), consistency index (CI), retention index (RI), rescaled consistency index (RC), and homoplasy index (HI)—were calculated for each maximum parsimonious tree generated. Multiple sequence alignment was also analyzed using ML in RAxML-HPC2 through the Cipres Science Gateway [55]. Branch support (BS) for ML analysis was determined by 1000 bootstrap replicates.

MrModeltest 2.3 [56] was used to determine the best-fit evolution model for each dataset for Bayesian inference (BI), which was performed using MrBayes 3.2.7a with a GTR+I+G model of DNA substitution and a gamma distribution rate variation across sites [57]. A total of 4 Markov chains were run, each consisting of 1.6 million generations, with random starting trees for 5.8S+nLSU (Figure 1) and 1.2 million generations for ITS (Figure 2) with trees and parameters sampled every 1000 generations. The first one-fourth of all generations were discarded as burn-in. The majority rule consensus tree of all remaining trees was calculated. Branches were considered as significantly supported if they received a maximum likelihood bootstrap value (BS) ≥ 70%, maximum parsimony bootstrap value (BT) ≥ 70%, or Bayesian posterior probabilities (BPP) ≥ 0.95.

## 3. Results

### 3.1. Molecular Phylogeny

The 5.8S+nLSU dataset (Figure 1) included sequences from 30 fungal samples representing 30 species. The dataset had an aligned length of 1508 characters, of which 1141 characters are constant, 104 are variable and parsimony uninformative, and 263 are parsimony informative. Maximum parsimony analysis yielded 54 equally parsimonious trees (TL = 986, CI = 0.5172, HI = 0.4828, RI = 0.5211, and RC = 0.2695). The best model was GTR+I+G (lset nst = 6, rates = invgamma; prset statefreqpr = dirichlet (1,1,1,1)). Bayesian and ML analyses showed a topology similar to that of MP analysis with split frequencies equal to 0.022581 (BI), and the effective sample size (ESS) across the two runs is double of the average ESS (avg ESS) = 869.5.

The 5.8S+nLSU rDNA gene regions (Figure 1) include ten genera within Trechisporales, *Brevicellicium*, *Dextrinocystis* Gilb. & M. Blackw., *Litschauerella* Oberw., *Luellia* K.H. Larss. & Hjortstam, *Scytinopogon*, *Sistotremastrum* J. Erikss., *Sphaerobasidium* Oberw., *Subulicystidium* Parmasto, *Tubulicium* Oberw., and *Trechispora*, shows that all related genera cluster into Trechisporales and the three new species grouped into *Trechispora*.

The ITS-alone dataset (Figure 2) included sequences from 42 fungal specimens representing 41 species. The dataset had an aligned length of 580 characters, of which 178 characters are constant, 61 are variable and parsimony-uninformative, and 341 are parsimony-informative. Maximum parsimony analysis yielded 584 equally parsimonious trees (TL = 2802, CI = 0.3123, HI = 0.6877, RI = 0.2519, and RC = 0.0787). Best model for the ITS dataset estimated and applied in the Bayesian analysis was GTR+I+G (lset nst = 6, rates = invgamma; prset statefreqpr = dirichlet (1,1,1,1). Bayesian and ML analyses resulted in a topology similar to that of MP analysis with split frequencies equal to 0.025000 (BI), and the effective sample size (ESS) across the two runs is double of the average ESS (avg ESS) = 621.5.

The phylogram inferred from the ITS dataset (Figure 2) indicated that the three new species grouped into *Trechispora*, in which the new species *T. murina* was sister to *T. bambusicola* with higher supports (96% BS, 92% BP, and 1.00 BPP); *T. odontioidea* formed a unique position within the clade of *T. fimbriata* C.L. Zhao and *T. nivea* (Pers.) K.H. Larss; while *T. olivacea* shared a clade formed by the members of *T. farinacea* (Pers.) Liberta, *T. hondurensis* Schoutteten & Haelew., and *T. mollis*.

### 3.2. Taxonomy

***Trechispora murina*** K.Y. Luo & C.L. Zhao, sp. nov. Figure 3 and Figure 4.

MycoBank no.: 842491.

**Holotype**—China, Yunnan Province, Wenshan, Funing County, Guying Village, GPS coordinates 23°44′ N, 105°56′ E, altitude 750 m asl., on a fallen angiosperm branch, leg. C.L. Zhao, 20 January 2019, CLZhao 11752 (SWFC).

**Etymology**—***murina*** (Lat.): Referring to the furry mouse-like hymenial surface.

**Basidiomata**—Annual, resupinate, thin, growing adnate but easily separable, up to 15 cm long, 3 cm wide, 100–500 µm thick. Hymenial surface grandinioid, pale greyish to grey when fresh, turn to greyish upon drying. Sterile margin concolorous with a hymenial surface, up to 2 mm wide.

**Hyphal system**—Monomitic; generative hyphae with clamp connections; colorless; thick-walled with a wide to lumen; richly branched; interwoven; encrusted; 2–3.5 µm in diameter; IKI–, CB–; tissues unchanged in KOH.

**Hymenium**—Cystidia and cystidioles absent; basidia more or less clavate, with four sterigmata and a basal clamp connection, 10.0–14.0 × 3.5–4.5 µm; basidioles dominant; basidioles in shape similar to basidia, but slightly smaller.

**Spores**—Basidiospores ellipsoid, colorless, thick-walled, ornamented, IKI–, CB–, (2.5–) 3–4 × (2–) 2.5–3 µm, L = 3.42 µm, W = 2.87 µm, Q = 1.17–1.20 (n = 60/2).

**Additional specimen examined (paratype)**—China, Yunnan Province, Wenshan, Funing County, Guying Village, GPS coordinates 23°39′ N, 105°59′ E, altitude 1400 m asl., on a fallen angiosperm branch, leg. C.L. Zhao, 20 January 2019, CLZhao 11736 (SWFC).

***Trechispora odontioidea*** K.Y. Luo & C.L. Zhao, sp. nov. Figure 5 and Figure 6.

MycoBank no.: 844493.

**Holotype**—China, Yunnan Province, Honghe, Pingbian County, Daweishan National Nature Reserve. GPS coordinates: 23°420′ N, 103°300′ E; altitude: 1500 m asl., on fallen angiosperm branches, leg. C.L. Zhao, 1 August 2019, CLZhao 17890 (SWFC).

**Etymology—*odontioidea*** (Lat.): Referring to the odontioid hymenophore.

**Basidiomata**—Annual, adnate, thin, up to 11 cm long, 2.5 cm wide, 50–200 µm thick. Hymenial surface odontioid, aculei cylindrical to conical, blunt, 0.3–0.6 mm long, pale buff when fresh, turn to buff upon drying. Sterile margin indistinct, cream to buff, 0.5–1 mm wide.

**Hyphal system**—Monomitic; generative hyphae with clamp connections; colorless, thin- to thick-walled; frequently branched; interwoven; 2–3.5 µm in diameter; ampullate hyphae frequently present; IKI−, CB−; tissues unchanged in KOH.

**Hymenium**—Cystidia and cystidioles absent.; basidia clavate, with four sterigmata and a basal clamp connection, 8.0–12.0 × 2.5–4 µm; basidioles dominant, in shape similar to basidia, but smaller.

**Spores**—Basidiospores subglobose to globose, colorless, slightly thick-walled, ornamented, IKI−, CB−, 2–3 × 1.5–2.5 µm, L = 2.53 µm, W = 2.00 µm, Q = 1.27 (n = 30/1).

***Trechispora olivacea*** K.Y. Luo & C.L. Zhao, sp. nov. Figure 7 and Figure 8.

MycoBank no.: 844494.

**Holotype****—**China, Yunnan Province, Honghe, Pingbian County, Daweishan National Nature Reserve. GPS coordinates: 23°420′ N, 103°300′ E; altitude: 1500 m asl., on fallen angiosperm branches, leg. C.L. Zhao, 1 August 2019, CLZhao 17826 (SWFC).

**Etymology****—*olivacea*** (Lat.): Referring to the olivaceous hymenial surface.

**Basidiomata**—Annual, resupinate, thin, very hard to separate from substrate, up to 11 cm long, 2.5 cm wide, 30–80 µm thick. Hymenial surface farinaceous, pale white to slightly olivaceous when fresh, turn to olivaceous upon drying. Sterile margin indistinct, slightly olivaceous, 0.2–0.5 mm wide.

**Hyphal system****—**Monomitic; generative hyphae with clamp connections; colorless; thin- to thick-walled; occasionally branched; interwoven; 1.5–3.0 µm in diameter; ampullate hyphae present; IKI–, CB–; tissues unchanged in KOH.

**Hymenium****—**Cystidia and cystidioles absent; basidia clavate, with four sterigmata and a basal clamp connection, 10.0–12.0 × 4.5–5 µm; basidioles dominant, with the shape similar to basidia, but smaller.

**Spores****—**Basidiospores broadly ellipsoid to globose, colorless, thick-walled, ornamented, IKI–, CB–, 2.5–4 × 1.5–2.5 µm, L = 3.30 µm, W = 2.65 µm, Q = 1.25 (n = 30/1).

## 4. Discussion

The classification of corticioid fungi revealed that two taxa of *Trechispora farinacea* and *T. hymenocystis* nested into *Trechispora* located in Hydnodontaceae (Trechisporales) [15]. In the present study (Figure 2), *Trechispora murina*, *T. odontioidea,* and *T. olivacea* are nested into *Trechispora*, in which *T. murina* was sister to *T. bambusicola*; *T. odontioidea* formed a monophyletic lineage and then grouped with *T. fimbriata* and *T. nivea*; while *T. olivacea* formed a monophyletic lineage and then grouped with *T. farinacea*, *T. hondurensis,* and *T. mollis*. However, *T. bambusicola* is morphologically distinguishable from *T. murina* by having the odontioid hymenophore with cream to buff the hymenial surface [33]. *Trechispora fimbriata* is distinguishable from *T. odontioidea* by having the hydnoid hymenial surface and longer basidiospores (3–3.6 × 2.4–3.2 µm) [33]; *T. nivea* differs from *T. odontioidea* by its thin-walled, larger basidiospores (3.5–4 × 2.5–3 µm) [5]. *Trechispora farinacea* is distinguishable from *T. olivacea* by its smooth to grandinioid or odontioid hymenophore with whitish to ochraceous hymenial surface and larger basidiospores (4–5 × 3.5–4 µm) [5]; *T. hondurensis* is separated from *T. olivacea* by having a hydnoid to partly irpicoid hymenial surface and thin-walled, wider basidiospores (3.6–3.8 × 2.7–2.9 µm) [58]; *T. mollis* is distinguishable from *T. olivacea* because it has white-yellow to pale yellow hydnoid hymenial surface, and wider ampullate septa at generative hyphae (reaching 8 µm in width) [26].

Morphologically, *Trechispora murina* is similar to *T. farinacea*, *T. rigida*, *T. subsphaerospora* (Litsch.) Liberta, and *T. torrendii* Chikowski & K.H. Larss. Based on the character of the grandinioid hymenial surface. However, *Trechispora farinacea* is separated from *T. murina* by having a whitish to ochraceous hymenial surface and larger, subglobose to broadly elliposid basidiospores (4–5 × 3.5–4 µm) [5]. *Trechispora rigida* differs from *T. murina* due to the presence of its dirty white to buff hymenophore [59] and having larger basidiospores (4.5–5.5 × 4 µm) [27]. *Trechispora subsphaerospora* differs from *T. murina* by having smooth basidiospores [34]. *Trechispora torrendii* differs in its pale yellow to yellow hymenophore [26] and has globose to subglobose basidiospores (2.8–3.5 × 3–3.5 µm) [27].

*Trechispora murina* is similar to *T. canariensis* Ryvarden & Liberta, *T. fastidiosa* (Pers.) Liberta, *T. praefocata* (Bourdot & Galzin) Liberta, *T. stevensonii* (Berk. & Broome) K.H. Larss., and *T. yunnanensis* C.L. Zhao due to the presence of the ellipsoid, ornamented basidiospores. However, *Trechispora canariensis* differs from *T. murina* because it has arachonoid to pelliculose hymenial surface and thin-walled, larger basidiospores (5–7 × 3–3.5 µm) [5]. *Trechispora fastidiosa* is separated from *T. murina* by having a membranaceous, whitish to cream hymenial surface and larger basidiospores (6–7 × 4.5–5.5 µm) [5]. *Trechispora praefocata* differs by having the farinaceous to arachnoid hymenial surface and larger basidiospores (5–6.5 × 4–5.5 µm) [5]. *Trechispora stevensonii* differs from *T. murina* by its hydnoid hymenophore and larger basidiospores (4–4.5 × 3–3.5 µm) [5]. *Trechispora yunnanensis* is separated from *T. murina* by having the farinaceous hymenial surface and larger basidiospores (7–8.5 × 5–5.5 µm) [31].

*Trechispora odontioidea* is similar to *T. bambusicola* C.L. Zhao and *T. nivea* in having an odontioid hymenial surface. However, *Trechispora bambusicola* differs from *T. odontioidea* because it has granulose basidiomata, and the absence of the ampullaceous septa [33]. *Trechispora nivea* differs from *T. odontioidea* due to the presence of white to ochraceous basidiomata and broadly ellipsoid to subglobose, thin-walled, larger basidiospores (3.5–4 × 2.5–3 µm) [5].

*Trechispora odontioidea* resembles *T. clancularis* (Park.-Rhodes) K.H. Larss., *T. cyatheae* Ordynets, Langer & K.H. Larss., *T. hymenocystis* (Berk. & Broome) K.H. Larsson, *T. invisitata* (H.S. Jacks.) Liberta, and *T. torrendii* Chikowski & K.H. Larss. due to the presence of ornamented or aculeate basidiospores. However, *Trechispora clancularis* is distinguishable from *T. odontioidea* due to the presence of its poroid to irpicoid hymenial surface and subglobose to ovoid, larger basidiospores (6–6.5 × 5–6 µm) [5]. *Trechispora cyatheae* differs from *T. odontioidea* in having a farinaceous to grandinioid hymenial surface, and broadly elliptical to slightly lacrymiform and adaxial side convex or straight, longer basidiospores (3–3.5 × 2–3 µm) [24]. *Trechispora hymenocystis* is distinguishable from *T. odontioidea* by its poroid basidiomata and broadly ellipsoidal to ellipsoidal, larger basidiospores (4.5–5.5 × 3.5–4.5 µm) [19]. *Trechispora invisitata* differs from *T. odontioidea* because it has a smooth to porulose, farinaceous to granulose hymenial surface and ellipsoid to ovate, larger basidiospores (4.5–5.5 × 3–4 µm) [5]. *Trechispora torrendii* differs from *T. odontioidea* because it has a farinose to grandinioid hymenial surface and larger basidiospores (3.2–3.5 × 2.8–3.2 µm) [26].

*Trechispora olivacea* is similar to *T. caucasica* (Parmasto) Liberta, *T. dimitica* Hallenb., *T. gelatinosa* Meiras-Ottoni & Gibertoni, *T. verruculosa* (G. Cunn.) K.H. Larss., and *T. yunnanensis* C.L. Zhao due to the presence of a farinaceous hymenial surface. However, *Trechispora caucasica* differs from *T. olivacea* by having a white to greyish hymenial surface and narrowly ellipsoid to reniform with a median constriction, larger basidiospores (5–5.5 × 4–4.5 µm) [5]. *Trechispora dimitica* differs from *T. olivacea* in its white to pale greyish hymenial surface, dimitic hyphal system, and shorter basidia (7–10 × 4.5–5.5 µm) [5]. *Trechispora gelatinosa* is distinguishable from *T. olivacea* by its coralloid basidiomata and wider basidiospores (3.2–4.5 × 2.5–3.5 µm) [27]. *Trechispora verruculosa* differs from *T. olivacea* because it has granulose to hydnoid with small cylindrical aculei, white to yellowish to ochraceous hymenial surface and larger basidiospores (4.5–5.5 × 3.5–4.5 µm) [5]. *Trechispora yunnanensis* can be delimited from *T. olivacea* by its larger basidiospores (7–8.5 × 5–5.5 µm) [31].

*Trechispora olivacea* resembles *T. hypogeton* (Maas Geest.) Hjortstam & K.H. Larss., *T. nivea*, *T. rigida,* and *T. thelephora* (Lév.) Ryvarden in having broadly ellipsoid to globose, ornamented basidiospores. However, *Trechispora hypogeton* is distinguishable from *T. olivacea* by its stipitate basidiomata and wider basidiospores (3.8–4.3 × 2.7–3.1 µm) [26]. *Trechispora nivea* differs from *T. olivacea* by the presence of a odontioid hymenial surface with white to pale ochraceous and wider basidiospores (3.5–4 × 2.5–3 µm) [5]. *Trechispora rigida* differs from *T. olivacea* due to the presence of a colliculose hymenial surface and larger basidiospores (4.5–5.5 × 4 µm) [26]. *Trechispora thelephora* differs from *T. olivacea* because it has a stipitate basidiomata and larger basidiospores (4.0–5.0 × 3.4–4.5 µm) [26].

Wood-rotting fungi are an extensively studied group of Basidiomycota [12,13,60,61,62,63,64,65,66] and the three taxa of *Trechispora* are a typical example group of wood-rotting fungi [15,33,34,35,67]. Based on our present morphology and phylogeny focusing on *Trechispora*, all taxa in this genus can be separated from the three new species.


**Key to 21 accepted species of *Trechispora* in China**


1. Basidiospores smooth-------------------------------------------------------------------------------------21′ Basidiospores aculeate, verrucose or ornamented--------------------------------------------------52. Ampullate hyphae > 5 μm in width, basidiospores angular----------------*T. subsphaerospora*2′ Ampullate hyphae < 5 μm in width, basidiospores ellipsoid------------------------------------33. Basidiospores thick-walled-----------------------------------------------------------------*T. cohaerens*3′ Basidiospores thin-walled--------------------------------------------------------------------------------44. Hymenial surface tuberculate-------------------------------------------------------*T. daweishanensis*4′ Hymenial surface smooth----------------------------------------------------------------------*T. xantha*5. Hyphal system dimitic------------------------------------------------------------------------*T. dimitica*5′ Hyphal system monomitic-------------------------------------------------------------------------------66. Hyphae without ampullate septa----------------------------------------------------------------------76′ Hyphae with ampullate septa-------------------------------------------------------------------------127. Basidiospores thin-walled, ovoid to subglobose---------------------------------------*T. suberosa*7′ Basidiospores thick-walled, ellipsoid-----------------------------------------------------------------88. Basidiospores > 7 μm in length--------------------------------------------------------*T. yunnanensis*8′ Basidiospores < 7 μm in length-------------------------------------------------------------------------99. Basidiomata margin greyish----------------------------------------------------------------***T. murina***9′ Basidiomata margin white to cream----------------------------------------------------------------1010. Hymenial surface odontioid---------------------------------------------------------*T. bambusicola*10′ Hymenial surface hydnoid--------------------------------------------------------------------------1111. Hymenophore with blunt aculei--------------------------------------------------------*T. fimbriata*11′ Hymenophore with sharp aculei--------------------------------------------------------*T. fissurata*12. Sphaerocysts present, hyphae inflated-------------------------------------------*T. hymenocystis*12′ Sphaerocysts absent, hyphae uninflated---------------------------------------------------------1313. Ampullate septa > 6 μm in width------------------------------------------------------------------1413′ Ampullate septa < 6 μm in width------------------------------------------------------------------1514. Basidiospores sparsely verrucose-----------------------------------------------*T. polygonospora*14′ Basidiospores densely aculeate---------------------------------------------------------*T. mollusca*15. Subhymenium with short-celled hyphae--------------------------------------------------------1615′ Subhymenium with long-celled hyphae---------------------------------------------------------1716. Basidiome thin, ochraceous--------------------------------------------------------------*T. farinacea*16′ Basidiome thick, dirty white to buff-------------------------------------------------------*T. rigida*17. Basidiospores thin-walled---------------------------------------------------------------------------1817′ Basidiospores thick-walled--------------------------------------------------------------------------1918. Hymenophore with hydnoid-----------------------------------------------------------------*T. nivea*18′ Hymenophore without hydnoid------------------------------------------------------*T. microspora*19. Basidiospores > 5 μm in length---------------------------------------------------------*T. praefocata*19′ Basidiospores < 5 μm in length---------------------------------------------------------------------2020. Hymenial surface farinaceous with olivaceous--------------------------------------***T. olivacea***20′ Hymenial surface odontioid with buff--------------------------------------------***T. odontioidea***

## Figures and Tables

**Figure 1 jof-08-01020-f001:**
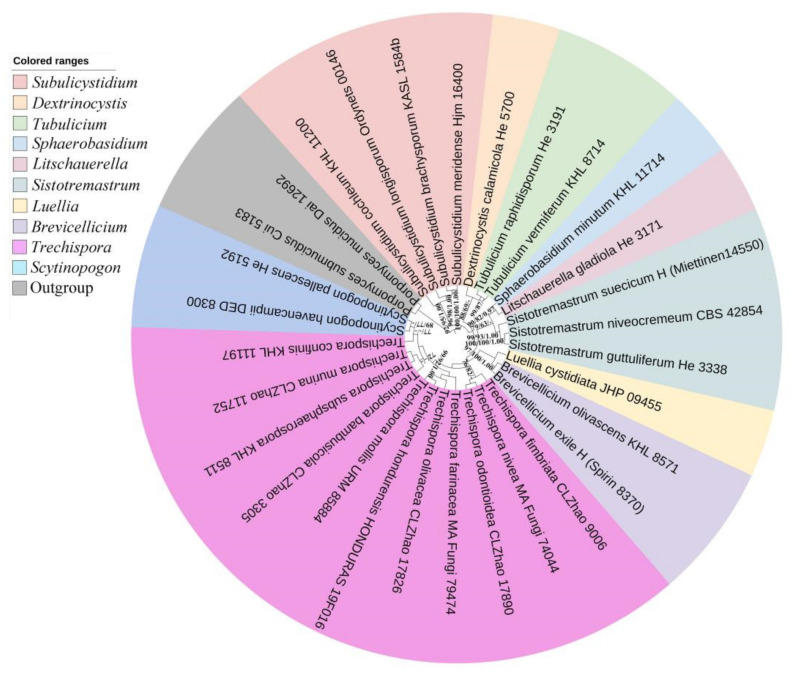
Maximum parsimony strict consensus tree illustrating the phylogeny of *Trechispora* and related genera in Trechisporales based on 5.8S+nLSU sequences. The genera represented by each color are indicated in the upper left of the phylogenetic tree. Branches are labelled with maximum likelihood bootstrap value ≧ 70%, parsimony bootstrap value ≧ 50%, and Bayesian posterior probabilities ≧ 0.95, respectively.

**Figure 2 jof-08-01020-f002:**
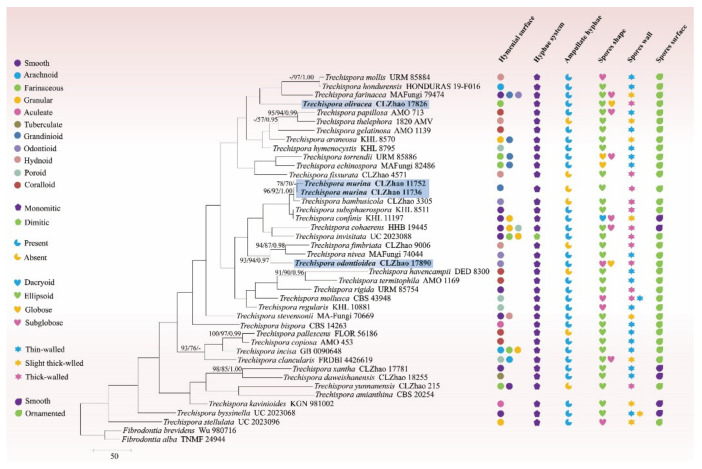
Maximum parsimony strict consensus tree illustrating the phylogeny of three new species and related species in *Trechispora* based on ITS sequences. Branches are labelled with maximum likelihood bootstrap value ≧ 70%, parsimony bootstrap value ≧ 50%, and Bayesian posterior probabilities ≧ 0.95, respectively. The new species are in bold.

**Figure 3 jof-08-01020-f003:**
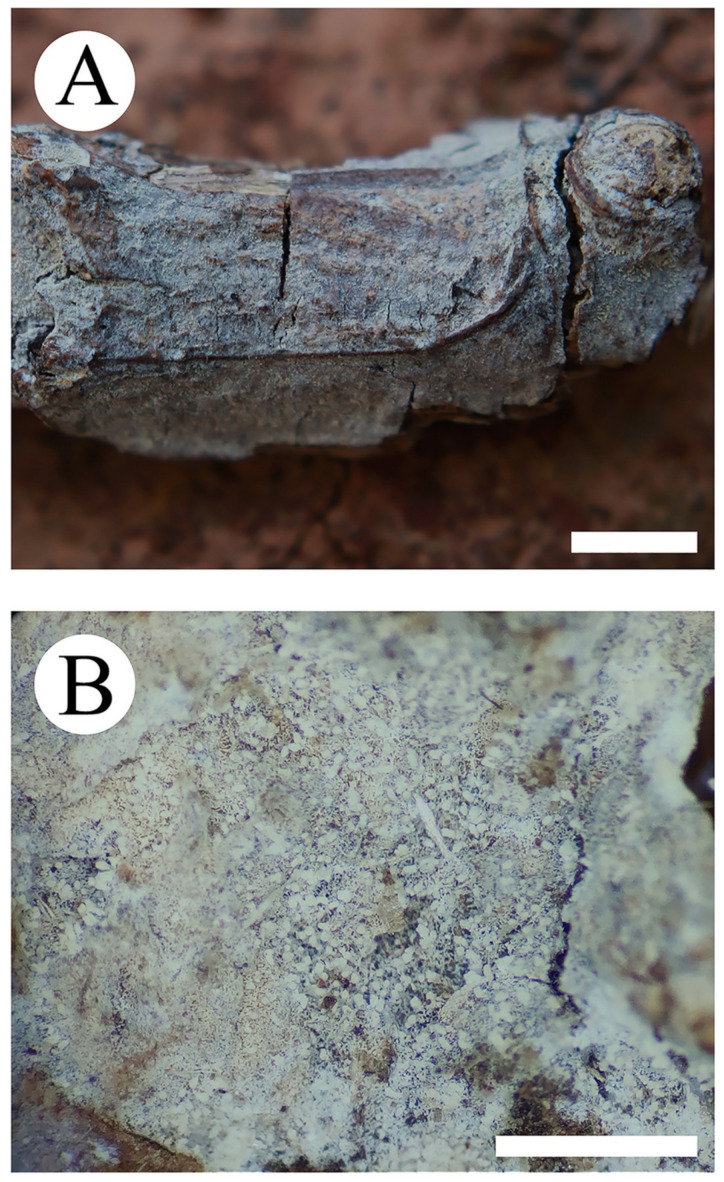
Basidiomata of *Trechispora murina* (holotype CLZhao 11752): the front of the basidiomata (**A**), characteristic hymenophore (**B**). Bars: (**A**) = 5 mm and (**B**) = 1 mm.

**Figure 4 jof-08-01020-f004:**
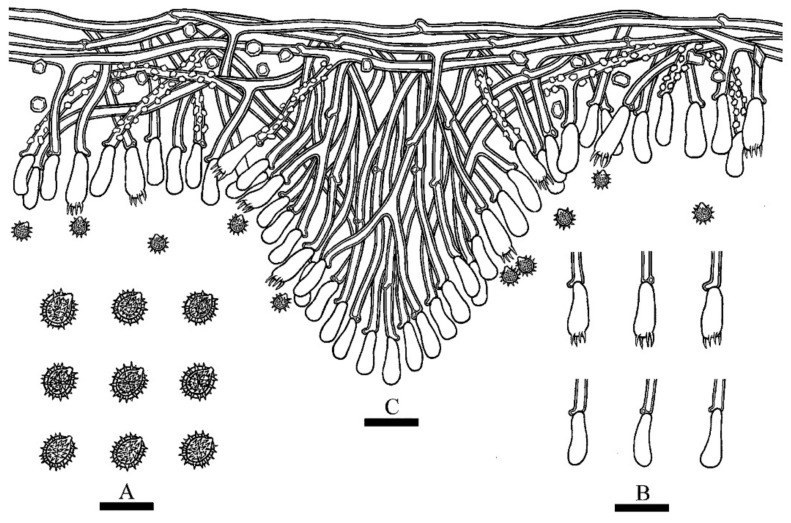
Microscopic structures of *Trechispora murina* (holotype CLZhao 11752): basidiospores (**A**), a cross-section of basidiomata (**B**), basidia and basidioles (**C**). Bars: (**A**) = 5 μm, (**B**,**C**) = 10 µm.

**Figure 5 jof-08-01020-f005:**
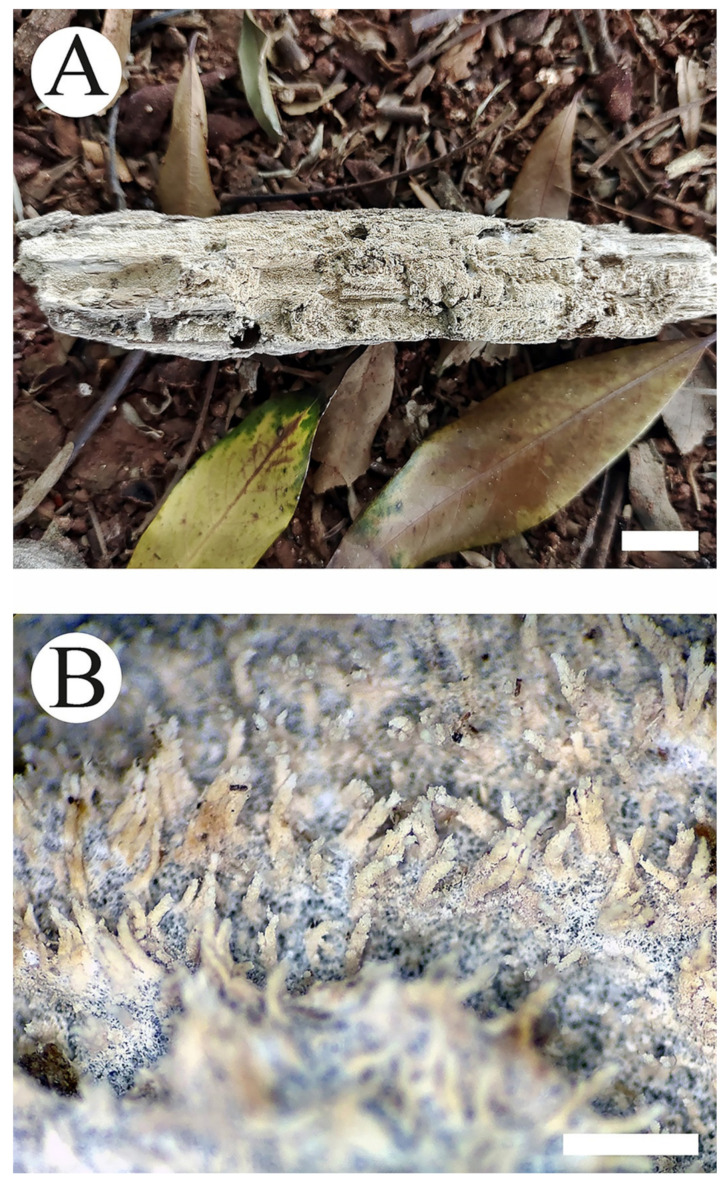
Basidiomata of *Trechispora odontioidea* (holotype CLZhao 17890): the front of the basidiomata (**A**), characteristic hymenophore (**B**). Bars: (**A**) = 1 cm and (**B**) = 1 mm.

**Figure 6 jof-08-01020-f006:**
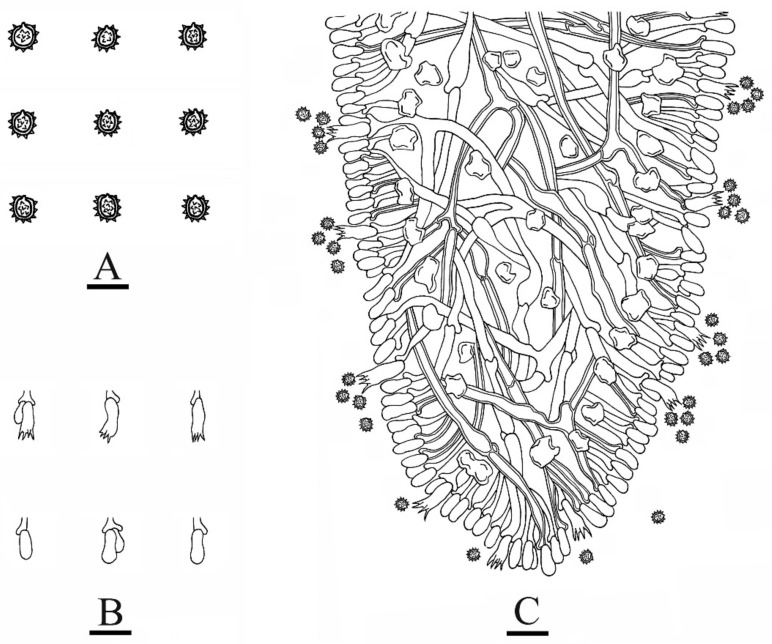
Microscopic structures of *Trechispora odontioidea* (holotype CLZhao 17890): basidiospores (**A**), basidia and basidioles (**B**), a cross section of basidiomata (**C**). Bars: (**A**) = 5 μm, (**B**,**C**) = 10 µm.

**Figure 7 jof-08-01020-f007:**
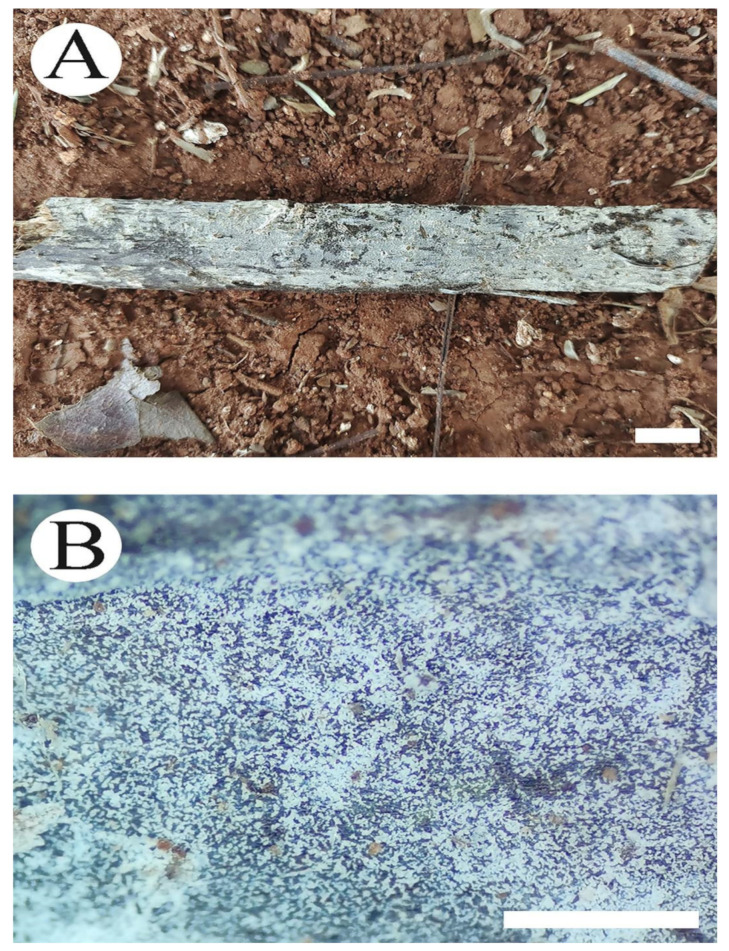
Basidiomata of *Trechispora olivacea* (holotype CLZhao 17826): the front of the basidiomata (**A**), characteristic hymenophore (**B**). Bars: (**A**) = 1 cm and (**B**) = 1 mm.

**Figure 8 jof-08-01020-f008:**
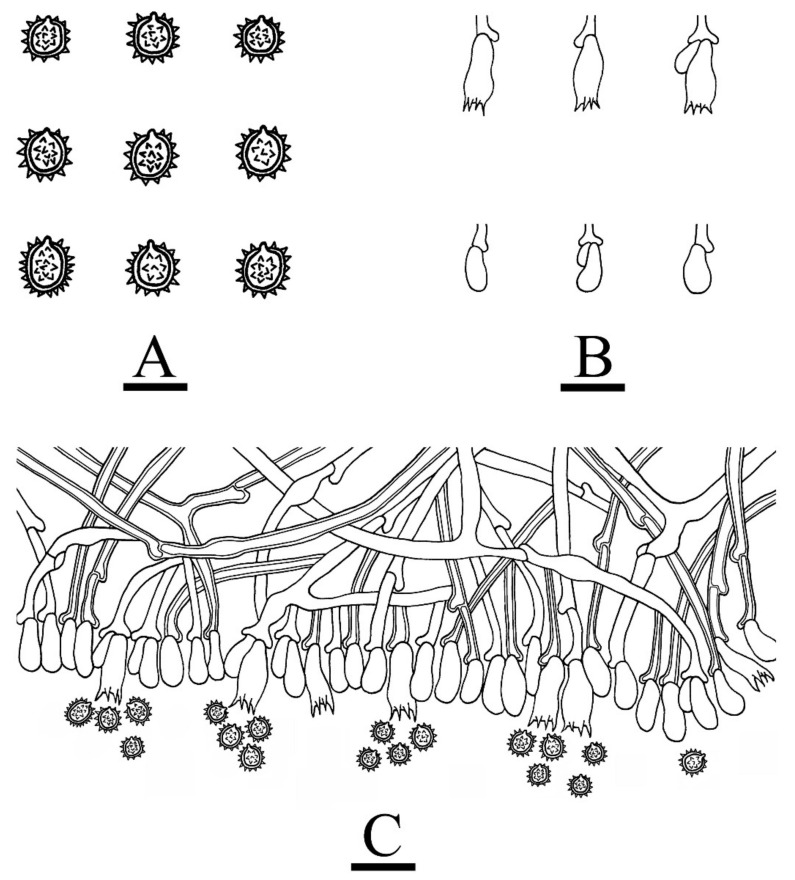
Microscopic structures of *Trechispora olivacea* (holotype CLZhao 17826): basidiospores (**A**), basidia and basidioles (**B**), a cross-section of basidiomata (**C**). Bars: (**A**) = 5 μm, (**B**,**C**) = 10 µm.

**Table 1 jof-08-01020-t001:** List of species, specimens, and GenBank accession numbers of sequences used in this study, the newly generated sequences are in bold fonts.

Species Name	Specimen No.	GenBank Accession No.		References
		ITS	nLSU	
*Brevicellicium exile*	H (Spirin 8370)	MT002322	MT002338	[43]
*B. olivascens*	KHL 8571	HE963792	HE963793	[36]
*Dextrinocystis calamicola*	He 5700	MK204534	MK204547	[38]
*Fibrodontia alba*	TNMF 24944	NR153983	NG060401	[24]
*F. brevidens*	Wu 9807-16	KC928276	KC928277	[44]
*Litschauerella gladiola*	He 3171	MK204555	MK204556	[38]
*Luellia cystidiata*	JHP 09455	MW371211		Unpublished
*Porpomyces mucidus*	Dai 12692	KT157833	KT157838	[45]
*P. submucidus*	Cui 5183	KT152143	KT152145	[45]
*Scytinopogon pallescens*	He 5192		MK204553	[38]
*S. havencampii*	DED 8300	KT253946	KT253947	[46]
*Sistotremastrum guttuliferum*	He 3338	MK204540	MK204552	[38]
*S. niveocremeum*	CBS 42854	MH857381	MH868921	[47]
*S. suecicum*	H (Miettinen14550)	MT075860	MT002336	[43]
*Sphaerobasidium minutum*	KHL 11714	DQ873652	DQ873653	[48]
*Subulicystidium brachysporum*	KASL 1584b	MH041544	MH041610	[49]
*S. cochleum*	KHL 11200	MN207036	MN207024	[50]
*S. longisporum*	Ordynets 00146	MN207039	MN207032	[50]
*S. meridense*	Hjm 16400	MH041538	MH041604	[49]
*Trechispora amianthina*	CBS 202.54	MH857292		[47]
*T. araneosa*	KHL 8570	AF347084		[35]
*T. bambusicola*	CLZhao 3305	MW544022	MW520172	[33]
*T. bispora*	CBS 142.63	MH858241		[47]
*T. byssinella*	UC 2023068	KP814481		Unpublished
*T. clancularis*	FRDBI 4426619	MW487976		Unpublished
*T. cohaerens*	HHB 19445	MW740327		Unpublished
*T. copiosa*	AMO 453	MN701018		[27]
*T. confinis*	KHL 11197	AY463473	AY586719	[35]
*T. daweishanensis*	CLZhao 18255	MW302338		[34]
*T. echinospora*	MA Fungi 82486	JX392853		[36]
*T. farinacea*	MA Fungi 79474	JX392855	JX392856	[36]
*T. fimbriata*	CLZhao 9006	MW544025	MW520175	[33]
*T. fissurata*	CLZhao 4571	MW544027		[33]
*T. gelatinosa*	AMO 1139	MN701021		[27]
*T. havencampii*	DED 8300	NR154418		[46]
*T. hondurensis*	HONDURAS 19-F016	MT571523	MT636540	Unpublished
*T. hymenocystis*	KHL 8795	AF347090		[35]
*T. incisa*	GB 0090648	KU747095		Unpublished
*T. invisitata*	UC 2023088	KP814425		Unpublished
*T. kavinioides*	KGN 981002	AF347086		[35]
*T. mollis*	URM 85884	MK514945		[26]
*T. mollusca*	CBS 43948	MH856428		[47]
** *T. murina* **	**CLZhao 11736**	**OL615003**		**Present study**
** *T. murina* **	**CLZhao 11752**	**OL615004**	**OL615009**	**Present study**
*T. nivea*	MA Fungi 74044	JX392832		[36]
** *T.* ** ** *odontioidea* **	**CLZhao 17890**	**ON417458**		**Present study**
** *T.* ** ** *olivacea* **	**CLZhao 17826**	**ON417457**		**Present study**
*T. pallescens*	FLOR 56186	MK458766		Unpublished
*T. papillosa*	AMO 713	MN701022		[27]
*T. regularis*	KHL 10881	AF347087		[35]
*T. rigida*	URM 85754	MT406381		[26]
*T. stellulata*	UC 2023096	KP814450		Unpublished
*T. stevensonii*	MA Fungi 70669	JX392841		[36]
*T. subsphaerospora*	KHL 8511	AF347080		[35]
*T. termitophila*	AMO 1169	MN701028		[27]
*T. thelephora*	1820 AMV	KF937369		[51]
*T. torrendii*	URM 85886	MK515148		[26]
*T. xantha*	CLZhao 17781	MW302340		[34]
*T. yunnanensis*	CLZhao 215	MN654923		[31]
*Tubulicium raphidisporum*	He 3191	MK204537	MK204545	[38]
*T. vermiferum*	KHL 8714	—	AY463477	[35]

## Data Availability

Publicly available datasets were analyzed in this study. This data can be found here: [https://www.ncbi.nlm.nih.gov/, accessed on 28 November 2021; https://www.mycobank.org/page/Simple%20names%20search].

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
