# Peer review of "A Molecular Systematics and Taxonomy Research on Trechispora (Hydnodontaceae, Trechisporales): Concentrating on Three New Trechispora Species from East Asia"

_jof, 2022, doi:10.3390/jof8101020_

Round 1

Reviewer 1 Report

Review report

Journal of Fungi (MDPI)

Paper No. jof-1900459

Paper title: A Molecular Mystematics and Taxonomy Research on Trechispora Hydnodontaceae, Trechisporales): Concentrating on Three New Trechispora Species from East Asia

The paper is well-written, with some amazing line drawings and phylogenetic tree presentation. I revised the abstract and made some other corrections in the text. After consulting these changes I recommend the paper for publication in Journal of Fungi.

Revise abstract: Trechispora is an important genus of wood-inhabiting fungi, have the ability to decompose the rotten wood in the forest ecosystem. In this study, we reported three new species of Trechispora: T. murina, T. odontioidea, T. olivacea from a subtropical region of Yunnan Province, China. Species descriptions were based on a combination of morphological features and phylogenetic analyses of ITS and LSU region of nuclear ribosomal DNA. Trechispora murina is characterized by the resupinate basidiomata, grandinioid hymenial surface with greyish tint, monomitic hyphal system and ellipsoid, thick-walled, ornamented basidiospores; T. odontioidea has an odontioid hymenial surface with cylindrical to conical, blunt aculei and subglobose to globose, colorless, slightly thick-walled, ornamented basidiospores; T. olivacea has a farinaceous hymenial surface with olivaceous tint, basidia clavate and thick-walled, ornamented, broadly ellipsoid to globose basidiospores. Sequences of the ITS and nLSU rDNA markers of the studied samples were generated, and phylogenetic analyses were performed with maximum likelihood, maximum parsimony, and Bayesian inference methods. After a series of phylogenetic analyses, the 5.8S+nLSU dataset was constructed to test the phylogenetic relationship of Trechispora with other genera of Hydnodontaceae. The ITS dataset was used to evaluate the phylogenetic relationship of the three new species with other species of Trechispora. Using ITS phylogeny, the new species T. murina was retrieved as a sister to T. bambusicola with moderate supports; T. odontioidea formed a single lineage and then grouped with T. fimbriata and T. nivea; while T. olivacea formed a monophyletic lineage with T. farinacea, T. hondurensis and T. mollis.

P2, L48, ‘infraspecifific’ replace with ‘intraspecific’

P2, L48, delete the word ‘individually’

P2, L50, replace “Trechispora’’ with ‘the genus’

P2, L52, replace ‘the molecular evidence’ with ‘nuclear ribosomal DNA sequence data’

P2, L52, replace ‘employed’ with ‘has been indicated’

P2, L53, replace ‘homobasidiomycetes’ with ‘Agaricomycetes’. Homobasidiomycetes are now a days not cited in literature.

P2, L55, delete the sentence ‘and the result did not point the morphological, physiological or ecological characters for Trechispora and related genera’.

P2, L57, 58, Please indicate the similar morphological characters of Trechispora to these genera.

P2, L58-61, delete this sentence ‘Based on the ITS and nLSU datasets, Ordynets et al. (2015) showed the phylogenetic study of Trechispora to report two new Trechispora species: T. cyatheae Ordynets, Langer & K.H. Larss. and T. echinocristallina Ordynets, Langer & K.H. Larss., which were found in La Réunion Island’. It showed be moved to the next paragraph.

P2, L65-73, replace the whole paragraph with ‘recently new species of Trechispora has been reported from North America and China (please add the references)

P2, L74, delete the word ‘recently’

P2, L74, replace ‘on’ with ‘of’

P2, L74, add after ‘fungi’ “Yunnan Province, China”

P2, L74, delete the word ‘additional’

P2, L75, delete the ‘from Yunnan Province, China’

P2, L81, please add the year after ‘collected’ like ‘during 2022’

P3, L120, in Table 1, add the words after ‘this study’ “the newly generated sequences are in bold fonts”

P3, Table1, replace regular fonts of the new species to bold fonts

P4, L123, replace ‘position’ with ‘to know the phylogenetic relationship of the’

P4, L124, delete ‘-only’, replace ‘datasets’ with ‘dataset’

P4, L124, replace the sentence ‘to position three new species among the Trechispora-related taxa’ with ‘to evaluate the phylogenetic relationships of the new species with known species of the genus’

P4, L130, “The three combined datasets” actually there are two datasets, please correct

P4, L132, replace ‘and the tree’ with ‘Maximum parsimony analyses’

P5, L145-146, replace the sentence ‘were run for 2 runs from random starting trees for 1.6 million generations’ with ‘were ran, each consisted of 1.6 million generation, with random starting trees’

P5, L179, replace ‘sequences analysis’ with ‘dataset’

P5, L182, replace ‘a single lineage and then grouped with’ with ‘a unique position within the clade of’

P5, L183, replace ‘formed a monophyletic lineage and then grouped with’ with ‘shared a clade forming by the members of’

P6, Figure 1. It will be better to present the regular tree instead up circular tree. This kind of tree is usually presented in a case with large number of sequences.

P7, L202, Figure 3. Please provide the holotype number (holotype xxxxx)

P8, L205, on fallen angiosperm branch? Please mentioned the name of tree(s).

P8, L225, Figure 4, again holotype number is missing

P8, L231, again the host’s plant name is missing

P9, Figure 5, add the holotype number

P10, Figure 6 add the holotype number

P10, L255, please mentioned the name of tree

P11, Figure, add holotype number

P12, delete L279 to 288.

Author Response

Reviewer #1:

The paper is well-written, with some amazing line drawings and phylogenetic tree presentation. I revised the abstract and made some other corrections in the text. After consulting these changes I recommend the paper for publication in Journal of Fungi.

Revise abstract: Trechispora is an important genus of wood-inhabiting fungi, have the ability to decompose the rotten wood in the forest ecosystem. In this study, we reported three new species of Trechispora: T. murina, T. odontioidea, T. olivacea from a subtropical region of Yunnan Province, China. Species descriptions were based on a combination of morphological features and phylogenetic analyses of ITS and LSU region of nuclear ribosomal DNA. Trechispora murina is characterized by the resupinate basidiomata, grandinioid hymenial surface with greyish tint, monomitic hyphal system and ellipsoid, thick-walled, ornamented basidiospores; T. odontioidea has an odontioid hymenial surface with cylindrical to conical, blunt aculei and subglobose to globose, colorless, slightly thick-walled, ornamented basidiospores; T. olivacea has a farinaceous hymenial surface with olivaceous tint, basidia clavate and thick-walled, ornamented, broadly ellipsoid to globose basidiospores. Sequences of the ITS and nLSU rDNA markers of the studied samples were generated, and phylogenetic analyses were performed with maximum likelihood, maximum parsimony, and Bayesian inference methods. After a series of phylogenetic analyses, the 5.8S+nLSU dataset was constructed to test the phylogenetic relationship of Trechispora with other genera of Hydnodontaceae. The ITS dataset was used to evaluate the phylogenetic relationship of the three new species with other species of Trechispora. Using ITS phylogeny, the new species T. murina was retrieved as a sister to T. bambusicola with moderate supports; T. odontioidea formed a single lineage and then grouped with T. fimbriata and T. nivea; while T. olivacea formed a monophyletic lineage with T. farinacea, T. hondurensis and T. mollis.

Response: Thank you for your patient revision. We agree with the abstract revision.

Introduction

1) P2, L48, ‘infraspecifific’ replace with ‘intraspecific’;

Response: We have revised it according to the reviewer’s comment.

2) P2, L49, delete the word ‘individually’;

Response: We have deleted it.

3) P2, L50, replace ‘Trechispora’ with ‘the genus’;

Response: We have replaced it according to the reviewer’s comment.

4) P2, L51, replace ‘the molecular evidence’ with ‘nuclear ribosomal DNA sequence data’;

Response: We have deleted the sentence.

5) P2, L52, replace ‘employed’ with ‘has been indicated’;

Response: We have revised the sentence’s kind of citation according to the reviewer’s comment.

6) P2, L51, replace ‘homobasidiomycetes’ with ‘Agaricomycetes’. Homobasidiomycetes are now a days not cited in literature;

Response: We have revised it.

7) P2, L54, delete the sentence ‘and the result did not point the morphological, physiological or ecological characters for Trechispora and related genera’;

Response: We have deleted it according to the reviewer’s comment.

8) P2, L55, Please indicate the similar morphological characters of Trechispora to these genera;

Response: We have added “, the similar morphological characters of Trechispora to these genera are basidiomata resupinate, hyphal system monomitic, cystidia absent.”

9) P2, L59-62, delete this sentence ‘Based on the ITS and nLSU datasets, Ordynets et al. (2015) showed the phylogenetic study of Trechispora to report two new Trechispora species: T. cyatheae Ordynets, Langer & K.H. Larss. and T. echinocristallina Ordynets, Langer & K.H. Larss., which were found in La Réunion Island’. It showed be moved to the next paragraph;

Response: We have moved the sentence to the next paragraph and revised the sentence’s kind of citation according to the reviewer’s comment.

10) P2, L64, replace the whole paragraph with ‘recently new species of Trechispora has been reported from North America and China’ (please add the references);

Response: We have replaced the whole paragraph and added the references.

11) P2, L66, delete the word ‘recently’;

Response: We have deleted it according to the reviewer’s comment.

12) P2, L66, replace ‘on’ with ‘of’;

Response: We have replaced it.

13) P2, L66, add after ‘fungi’ ‘Yunnan Province, China’;

Response: We have added it according to the reviewer’s comment.

14) P2, L67, delete the word ‘additional’;

Response: We have deleted it.

15) P2, L68, delete the ‘from Yunnan Province, China’;

Response: We have deleted it according to the reviewer’s comment.

Materials and Methods

1) P2, L73, please add the year after ‘collected’ like ‘during 2022’;

Response: We have added “during 2019” after “collected”.

2) P3, L113, in Table 1, add the words after ‘this study’ ‘the newly generated sequences are in bold fonts’;

Response: We have added it according to the reviewer’s comment.

3) P3, Table1, replace regular fonts of the new species to bold fonts;

Response: We have revised them.

4) P4, L117, replace ‘position’ with ‘to know the phylogenetic relationship of the’;

Response: We have replaced it according to the reviewer’s comment.

5) P4, L118, delete ‘-only’, replace ‘datasets’ with ‘dataset’;

Response: We have revised them.

6) P4, L118, replace the sentence ‘to position three new species among the Trechispora-related taxa’ with ‘to evaluate the phylogenetic relationships of the new species with known species of the genus’;

Response: We have replaced it according to the reviewer’s comment.

7) P4, L125, ‘The three combined datasets’ actually there are two datasets, please correct;

Response: We have confirmed there is “The three combined datasets”.

8) P4, L127, replace ‘and the tree’ with ‘Maximum parsimony analyses’;

Response: We have revised it.

9) P5, L141-142, replace the sentence ‘were run for 2 runs from random starting trees for 1.6 million generations’ with ‘were ran, each consisted of 1.6 million generation, with random starting trees’;

Response: We have revised it according to the reviewer’s comment.

Results

1) P5, L176, replace ‘sequences analysis’ with ‘dataset’;

Response: We have revised it.

2) P5, L179, replace ‘a single lineage and then grouped with’ with ‘a unique position within the clade of’;

Response: We have revised it according to the reviewer’s comment.

3) P5, L180, replace ‘formed a monophyletic lineage and then grouped with’ with ‘shared a clade forming by the members of’;

Response: We have revised it.

4) P6, Figure 1. It will be better to present the regular tree instead up circular tree. This kind of tree is usually presented in a case with large number of sequences;

Response: Thanks for your kindly and good suggestions. If it is possible, we would like to show it as the circular tree based on the interesting way to show to reader for this time, and we will show it as the regular tree instead up circular tree next time.

5) P7, L199, Figure 3. Please provide the holotype number (holotype xxxxx);

Response: We have added the holotype number in Figure 3.

6) P8, L202, on fallen angiosperm branch? Please mentioned the name of tree(s);

Response: We have checked it carefully and the host tree is being as the serious wood-rotting type, which is hard to identify the name for them.

7) P8, L221, Figure 4, again holotype number is missing;

Response: We have added the holotype number in Figure 4.

8) P8, L227, again the host’s plant name is missing;

Response: We have checked it carefully and the host tree is being as the serious wood-rotting type, which is hard to identify the name for them.

9) P9, Figure 5, add the holotype number;

Response: We have added the holotype number in Figure 5.

10) P10, Figure 6 add the holotype number;

Response: We have added the holotype number in Figure 6.

11) P10, L251, please mentioned the name of tree;

Response: We have checked it carefully and the host tree is being as the serious wood-rotting type, which is hard to identify the name for them.

12) P11, Figure 7, add holotype number;

Response: We have added the holotype number in Figures 7, 8.

Discussion

1) P12, delete L272 to 281.

Response: We have deleted it according to the reviewer’s comment.

Reviewer 2 Report

Dear authors

this is a well written taxonomic paper, congratulations.

nevertheless, some things have to be improoved:

There are 2 ways of citing literture. You have to check which is the correct citing mode for JFungi.

Fig 1 has to be rearranged so that the centre with the branches and the numbers can be anlysed without a microscope

In Fig 6 you are drawing the tip of a aculeus (not a cross section of a carpophore). You should recheck whether mature hymenium is really covering the complete tip and whether the arrangement of the hyphae is really as irregular as given. Normally the tip of the aculei is sterile and built up of regularly arragned hyphae.

Author Response

Dear reviewer,

This is a well written taxonomic paper, congratulations.

nevertheless, some things have to be improved:

1) There are 2 ways of citing literture. You have to check which is the correct citing mode for JFungi.

Response: We have checked citations literture and revised them.

2) Fig 1 has to be rearranged so that the centre with the branches and the numbers can be anlysed without a microscope.

Response: Thanks for your great comment. We tried to magnify the numbers and show them as bold type.

3) In Fig 6 you are drawing the tip of a aculeus (not a cross section of a carpophore). You should recheck whether mature hymenium is really covering the complete tip and whether the arrangement of the hyphae is really as irregular as given. Normally the tip of the aculei is sterile and built up of regularly arragned hyphae.

Response: We have checked it carefully, and the mature hymenium was selected to show the complete tip and the hyphae is really as irregular as given after checked again.

Title

1) line 2: revised “Mystematics” as “Systematics”;

Response: We have revised it according to the reviewer’s comment.

Introduction

1) line 51-52: This is trivial;

Response: We have deleted it.

2) line 51: you have 2 different kinds of citations: with numbers or with author and year; you have to check which is the correct way for JoF;

Response: We have revised them according to the reviewer’s comment.

Materials and methods

1) line 84: “Rayner (1970) and Petersen (1996)” why not a color chart like Kornerup & Wanscher?

Response: We have revised it as “Color terminology follows Kornerup and Wanscher”.

2) line 90: “tubes”?

Response: We have revised it as “basidiomata”.

3) line 96: does this mean that from each collection (a) number of spores was measured or does it mean that (a) number of spores were measured in total, coming from (b) number of specimen?

Response: We have revised it as “ (a) number of spores were measured in total, coming from (b) number of specimen.”

Results

1) Figure 1: The tree and the numbers are almost imposible to be seen - this should be improved!

Response: We tried to magnify the numbers and show them as bold type.

2) Figure 6: Is the hymenium really contiuously covering the aculei? The tips of the aculiei are normally sterile;

Response: We have checked it carefully, and the mature hymenium was selected to show the complete tip and the hyphae is really as irregular as given after checked again.

Discussion

1) line 286: revise “the” as “a”;

Response: We have revised it according to the reviewer’s comment.

2) line 288: delete “with”;

Response: We have deleted it.

3) line 288: add “at”;

Response: We have added it according to the reviewer’s comment.

4) line 297: delete “the”;

Response: We have deleted it.

5) line 314: delete “the”;

Response: We have deleted it according to the reviewer’s comment.

6) line 359: delete “of”.

Response: We have deleted it.